# Factors influencing place of delivery in Ethiopia: Linking individual, household, and health facility-level data

**Fanuel Belayneh Bekele**[1]*, **Kasiye Shiferaw**[2], **Adiam Nega**[3], **Anagaw Derseh**[3], **Assefa Seme**[3], **Solomon Shiferaw**[3]

1 School of Public Health, Hawassa University, Hawassa, Ethiopia, 2 School of Midwifery, Haromaya University, Haromaya, Ethiopia, 3 School of Public Health, Addis Ababa University, Addis Ababa, Ethiopia

* fanuelbelayneh@gmail.com

## Abstract

**Data Availability Statement:** The dataset is publicly available on https://www.pmadata.org/data/request-access-datasets. Anyone can access the dataset in the same manner as the authors by

### Introduction

Maternal mortality remains high, especially in sub-Saharan Africa. Institutional delivery is one of the key intervention to reduce it. Despite service utilization reflects an interplay of demand- and supply-side factors, previous studies mainly focused on either sides due to methodological challenges and data availability. But, a more comprehensive understanding can be obtained by assessing both sides. The aim of this study is to assess individual, household, community, and health facility factors associated with deliveryplace in Ethiopia.

### Methods

We have used the 2019 Performance Monitoring for Action survey data set, which is a nationally representative sample of women linked with national sample of health facilities in Ethiopia. A total of 2547 women who recently delivered were linked with 170 health centers and 41 hospitals. Facility readiness index was calculated based on previous study conducted by Stierman EK on similar data set. We applied survey weights for descriptive statistics. Multilevel mixed-effects logistic regression was used to identify factors influencing delivery place.

### Results

Coverage of institutional delivery was 54.49%. Women aged 20–34 [AOR; 0.55 (0.32–0.85)] compared with those younger than 20 years; those with no formal education [AOR: 0.19 (10.05–0.76)] or attended only primary school [AOR: 0.20 (0.05–0.75)] compared with those attended above secondary; and women whose partners didn't encourage antinatal visit [AOR; 0.57 (0.33–0.98)] all have decreased odd of institutional delivery. Attending at least one antenatal visit [AOR: 3.09 (1.87–5.10)] and increased availability of medicines in the closest facility [AOR: 17.33 (1.32–26.4)] increase odds of institutional deliver.

submitting an online application. We the authors also have no special access privileges for these data and it is publicly available for everyone.

**Funding:** The authors received no specific funding for this work.

**Competing interests:** The authors have declared that no competing interests exist.

## Conclusion

In Ethiopia, nearly half of the total deliveries take place outside health facilities. In addition to improving women's education, utilization of antenatal care, and encouragement by partners, it is important to consider the availability of medicine and commodities in the nearby health facilities while designing and implementing programs to reduce home delivery.

## Introduction

In the past two and half decades maternal mortality declined by 50% however, far behind to meet the Sustainable Development Goal 3 (SDG 3), target of less than 70 per 100,000 live births between 2016 and 2030 [1]. Currently, 830 women die every day from preventable causes related to pregnancy and childbirth, globally. An estimated 287,000 maternal deaths occur every year, 99% are from developing countries, with nearly half of these take place in Sub-Saharan Africa [2]. Ethiopia has a high Maternal Mortality Ratio (MMR) (412 deaths per 100,000 live births) which is about two times higher than the average global MMR (211 per 100,000 live births) [3,4].

The causes of maternal death are directly related to the high risk of being pregnant and the obstetric risk of developing complications. The major complications which account for 80% of all maternal deaths are hemorrhage, sepsis, hypertension, and unsafe abortion. Approximately 75% of maternal deaths are preventable if all women have access to interventions for managing pregnancy and preventing and treating birth complications. Therefore, institutional delivery is a key approch to reduce maternal mortality by reducing the incidence of complications related to pregnancy and childbirth [5,6].

In 2016, the global coverage of skilled birth attendance was 78%. However, about 40% of births in African countries were not assisted by a skilled birth attendant, and 74.7–89.9% of women gave birth at home in Sub Saharan Africa [2]. In Ethiopia, only 48% of deliveries were assisted by skilled birth attendant [7]. Other studies conducted in different parts of Ethiopia also reported that institutional delivery ranges from 27 to 51% [8–13].

A number of studies were also conducted to assess factors influencing place of delivery in Ethiopia. Majority of these attempts focus on identifying the demand-side factors, such as those occurring at the individual, household or community levels. These studies showed that living in urban areas, increase in education level of a woman, living in households with better economic status, increase in maternal age, attending ANC visit, knowing pregnancy danger signs, having birth preparedness, and having access to mobile phone and availability of radio or TV are factors that increase the level of institutional delivery [8–12].

However, the use of health services reflects the interplay of demand-side factors and supply-side factors like location of health facilities, availability of trained health workers, infrastructure, and supplies [14,15]. Due to lack of service provision data and methodological challenges, previous studies by separate analyses of the demand- and the supply-side factors offered limited insight. Most studies overlooked the supply-side factors and attempts of analyzing facility level factors by adjusting for the demand-side factors were barely applied in previous studies [8–12,16].

As an alternative for separate analyses on demand- and the supply-side factors, linking together information across multiple data sources from facility and community surveys has become an attractive approach [16,17]. Generally, there are two types of methods to establish links between survey respondents and individual facilities. The first one is linking based on

geographic proximity or respondent identification of facility [1] visited. Another approach links household clusters to all facilities within a geographic area. However, linking household with all facilities in the cluster or theoretical catchment creates a homogenous service environment within the boundary. This approach is preferred while using independently sampled surveys or similar type of facilities [18].

Previous studies which attempted to link data from Ethiopian Demographic and Health Survey (EDHS) with Provision Assessments (SPA) have faced several methodological limitations. One limitation of linking EDHS and SPA data is the fact that the surveys are rarely executed at the same time and within the same clusters, limiting the inferences to be made from the result of such analyses [13,19–21]. The other weakness related with cluster level linkage by previous studies was the misclassification bias introduced as a result of large variations in facility readiness [22].

We have tried to address the above limitations by linking household panel survey data with Service Delivery Point (SDP), or facility, survey data from Performance Monitoring for Action (PMA-Ethiopia). This survey assessed all public facilities in the selected Enumeration Areas (EAs) at the same time the elegible women in the area were enrolled in the panel survey. As a result, the health facilities included in the survey are those administratively assigned to provide service in the selected EAs; i.e. sampling of health facilities and the enumeration areas for the community survey were not independent [23]. By further linking households with the closest public health facility, we can minimize the misclassification bias that could arise due to cluster level linkage methods. In addition, our analysis was performed based on the assumption that delivering in a health facility is associated with the readiness of nearby facilities to provide quality delivery-related services, after adjusting for the community or household characteristics as well as for the mother's socioeconomic status, age, birth order, and marital status [22]. Further more, we assume that, use of health services reflects the interplay of demand- and supply-side factors. Based on this, individual, household and community level factors were not only considered for statistical adjustment but also assessed for their independent effect on place of delivery [14,15].

Therefore, this study analyzed the individual, household, community, and health facility level factors affecting place of delivery by linking the individual, household, and facility data from the PMA-Ethiopia 2019 survey.

## Method and materials

### Study design and setting

Performance Monitoring for Action Ethiopia (PMA Ethiopia) employed a panel study design to identify gaps in Reproductive, Maternal, and Newborn Health (RMNH) care among a nationally representative sample of pregnant and postpartum women and their infants in Ethiopia. Altogether, 265 Enumeration Areas (EAs), or geographic sampling units, were drawn separately from rural and urban strata within the Tigray, Amhara, Oromia, and South Nation, Nationalities and Peoples (SNNP), whereas randomly selected with probability proportional to size within Afar region without rural or urban stratification. All EAs were drawn from urban areas without stratification since Addis Ababa is exclusively urban.

### Data source

Data used in this analysis were collected between October 2019 to December 2020. We used the 2019 PMA panel baseline and six-weeks interview data sets linked with the Service SDP 2019 data set. Detailed PMA projects and data collection protocols have been reported elsewhere [23]. A multi-stage cluster sampling technique was used to draw a probability sample of

households and women of reproductive age. In the panel survey, eligible women (all pregnant or immediate postpartum women living in the EAs) were identified by using a screening question form. The eligible women were pregnants or currently less than 9 weeks postpartum women, stayed in the parents' home for pregnancy or postpartum period who reside in the EAs. All these eligible enrolled women were then interviewed at baseline and between five to eight weeks postpartum. The public and private facilities that serve the identified EAs were included in the SDP survey. All facilities identified as serving the enumeration area through the sampling process are eligible for participation. A list of all public and private health facilities from the local district health offices that included all health posts, health centers, and hospitals in corresponding districts was obtained, once EAs were identified. The list of all private health facilities in each *kebele* is reviewed to sample three private health facilities and all levels of public SDPs serving the selected EAs were included. A facility readiness was the focus of the survey for offering essential RMNH services, while also capturing additional provision of quality of care.

PMA uses mobile data collection technology for both community and facility surveys. The questionnaires were programmed using open-source software called Open Data Kit (ODK) for collecting and managing data. Data were collected by trained Resident Enumerators who have a minimum of diploma level of education. A group of REs was closely followed by assigned supervisors and Regional Coordinators (RCs). In addition to the intelligent checks employed at the design phase of data collection forms, there is also a central data management team assigned to follow the quality and progress of the data collection process.

## Variables and measurements

Our dependent variable was place of delivery, coded as 1 if a woman used a health facility for delivery care for the most recent birth and 0 if otherwise. The independent variables were grouped into individual-level, community and household, and health facility-level factors. Individual-level factors include: Age (<20, 20–34, 35–49); Marital status (Married or Others); Maternal education (Never attended school, Primary education, secondary education, technical or vocational, Higher education); ANC visit (Yes or No); Current pregnancy desired (Then, Later, Not at all); Birth events (Primipara, Multipara, Grandmultipara); Ever been pregnant (Yes or No); Use FP (Ever user or Never Used); Other pregnancy in the last 2 years (Yes or No); Ever deliver in a health facility (Yes or No); Seen HEW &/or other HP for ANC (Yes or No); and Current pregnancy desired (Then, Later, Not at all). Community and household factors include Residence (Urban or Rural); Community encourage facility delivery (Yes or No); Community encourage delivery by TBA (Yes or No); Family size (≤3, 4–6, ≥6); Discuss planned delivery place with a partner (Yes or No); and Wealth Index (Lowest, Lower, Middle, Higher, Highest).

The indicators used by PMA Ethiopia to measure obstetric and newborn care at health facilities were selected based on the recommended items by WHO standards for maternal and newborn for Service Availability and Readiness Assessment (SARA) [24,25]. In this study facility readiness index was calculated based on previous study conducted by Stierman EK on similar data set [26]. Based on this, the indicators were grouped into four domains; 1) medicines and commodities observed, 2) equipment, supplies, and amenities available, 3) performance of signal functions indicators and 4) Staffing and systems to support quality in the facilities. In these four domains, there were 52 indicators to assess hospitals and 44 indicators for health centers. A selected indicator was assigned a value of "1" if the item was available or if the service has been provided in the last three months, and, "0" otherwise. Finally, services readiness indexes were calculated as mean availability of items as a percentage within each domain.

The equal-weight approach, in which equal weight was given to each domain and each indicator within the same domain was used. Compared with other weighting approaches giving equal weight to all indicators and standardizing the sum to have a maximum value of 100% was a recommended approach for calculating composite indicators [27,28]. We calculated the readiness scores separately for hospitals and health centers considering the difference in the services provided at each level in the country. The details of the indicators in each domain were summarized suplementery document (S1 Table).

## Linking method

PMA-Ethiopia is an important new source of data for researchers in which household and service delivery point data are gathered simultaneously and can be geographically linked which will improve effective coverage measurement and address many of the limitations that hinder current research efforts. Accordingly, we linked the panel survey and SDP datasets from PMA-Ethiopia 2019 survey. First, we merged the Household/Female baseline and six weeks postpartum interview datasets. We restricted observations to females with completed interviews in both data sets. Then, we used Quantum Geographic Information System (QGIS) software to link every household to the closest SDP based on GPS locations. To do so, we used the Join by nearest tool in QGIS, which allows us to link two datasets based on the spatial relationship between data points. Specifically, we linked observations in the two datasets based on geographic proximity. The tool linked all observations from the merged household/female baseline and the follow-up dataset to the nearest observation from the SDP dataset using the shortest straight line (Euclidian) distance as the main linking criteria. Despite the limitations associated with this type of geographical linkage method, the use of PMA-Ethiopia panel and SDP data collected simultaneously as well as the protocol of the survey to select both panel women and facilities located within the EAs/*kebeles* (the lowest of administrative divisions in Ethiopia) can improve the effectiveness of the coverage measurement.

## Analysis

The analysis starts with descriptive statistics in three dimensions. First, individual, household and community level factors were described by using frequencies, percentages mean or standard deviation. Second, at the health facility level, we described the background characteristics of health facilities that provide the service, the availability of services, commodities, or supplies at these facilities.

Respondents who live in the same EA may not be independent of one another. Thus, the individual-level analysis ignores the nesting of people within clusters, which can result in the underestimation of standard errors. Moreover, the outcome variable is at the individual level but the key explanatory variables of most interest, the service environment indicators, are at the cluster level. Due to these, a multilevel analysis approach is more appropriate to allow for simultaneous investigation of the effects of the group-level and individual-level predictors on individual-level outcomes. Therefore, multilevel mixed-effects logistic regression was used to identify factors with binary dependent variables; institutional delivery (with a value of "1"), home delivery otherwise (with a value of "0"). The effect of a complex sample design was taken into account by applying survey weights in the descriptive analysis.

## Ethics

The survey was conducted after obtaining IRB approval from Addis Ababa University and Johns Hopkins School of Public Health. Publicly available PMA-Ethiopia datasets were

obtained by online application submitted to Johns Hopkins School of Public Health (via-https://www.pmadata.org/data/available-datasets/request-accessdatasets).

## Results

A total of 2547 women with complete information about their delivery place were included in this study. The mean or standard deviation of women's age was 27.18 (±6.34) years and the average number of household members was 4.84(±2.12 SD). Fourty percent of the women in this study have never attended school while the other 40% have primary level of education. Primipara women account for 458 (17.99%) of the respondents, whereas, 1,546 (60.72%) were multipara and 542 (21.29%) grand multipara. A total of 1,061 (51.59%) women have previous experience of institutional delivery. Women who had at least one antenatal visit were 2,062 (80.98%). The current pregnancy was based on their desire for 46.88% of the mothers (Table 1).

One thousand eight hundred sixty-one (77.96%) of mothers included in this study live in rural areas. There were 878 (34.47%) housholds with a family size between four and five. Regarding partners involvement; 2,135 (83.83) had partner who encourage ANC visit & 1,830 (71.87) pland their place of delivery with their partner. One thusand three hundred four (51.21%) reported that most people in their community encourage institutional delivery while 457 (17.94%) said most people in their community encourage to use traditional birth attendants (Table 2).

### Coverage of health facility delivery

Among total of 2,547 mothers, 1,571 (54.49%; 95% CI; 52.16%-56.82%) had health facility deliveries and 976 (45.51%; 95% CI; 43.17%-47.84%) had home deliveries. The lowest proportion of institutional delivery was observed in the Afar region (18.7%) while the highest was in Addis Ababa (97.1%). Amhara, Oromia, and SNNPR regions were having proportion facility delivery between 43–50% (Fig 1).

### Facility readiness for delivery services

A total of 46 (21.30%) public hospitals and 170 (78.70%) health centers providing delivery service with completed SDP results were included in this analysis. The overall weighted mean score (±SD) for all the four domains of the WHO standards childbirth readiness index; 1) medicines and commodities observed, 2) equipment, supplies, and amenities available, 3) performance of signal functions indicators and 4) Staffing and systems to support quality in the facilities was 74.28% (±9.95%) for hospitals and 63.17% (±11.40%) for health centers. For the first domain, on average hospitals have 35.58% (±9.12%) of the required medicines and commodities while health centers have 31.84% (±10.88%). Availability of equipment, supplies, and amenities used for delivery service were 85.94% (±15.24%) in hospitals and 77.46% (±17.39%) in health centers. The average performance of the signal function in the public health facilities was 65.63% (±5.95%). Staffing and required items to support quality in hospitals and health centers were available at the average level of 90.75% (±121.03%) and 84.93% (±12.56%) respectively (Table 3).

### Factors associated with place of delivery

The Intra Cluster Correlation (34.63%) indicate that clusters of EAs in the survey explain about 34% of the variance in place delivery. In the final multilevel mixed-effects logistic regression model, younger maternal age, higher level of maternal education, ever using modern

Table 1. Description of individual factors by place of delivery, PMA-Ethiopia 2019.

| Maternal characteristics & individual factors | Place of delivery | | | | p-value |
|---|---|---|---|---|---|
| | Home (976) | Facility (1571) | Total | | |
| | No (%) | No (%) | No (%) | | |
| **Mothers' age** | | | | | |
| <20 years | 175 (39.20) | 271 (60.80) | 446 (17.50) | | <0.001 |
| 20–34 years | 756 (44.68) | 937 (55.32) | 1,693 (66.49) | | |
| 35–49 years | 228 (55.85) | 180 (44.15) | 408 (16.01) | | |
| **Marital status** | | | | | |
| Married | 1,115 (45.90) | 1,314 (54.10) | 2,428 (95.34) | | 0.002 |
| Living together | 22 (33.70) | 43 (66.30) | 65 (2.56) | | |
| Divorced/Widowed | 18 (43.52) | 23 (56.48) | 41 (1.60) | | |
| Never Married | 5(36.75) | 8(63.25) | 13(0.50) | | |
| **Maternal education** | | | | | |
| Never attended school | 642 (61.89) | 395 (38.11) | 1,038 (40.75) | | <0.001 |
| Primary | 451 (44.04) | 573 (55.96) | 1,024 (40.19) | | |
| Secondary | 57 (19.14) | 243 (80.86) | 300 (11.80) | | |
| More than secondary | 8 (4.55) | 176 (54.49) | 185 (7.26) | | |
| **Parity** | | | | | |
| Primipara | 100 (21.90) | 358 (78.10) | 458 (17.99) | | <0.001 |
| Multipara | 707 (45.70) | 840 (54.30) | 1,546 (60.72) | | |
| Grand multipara | 352 (64.90) | 190 (35.10) | 542 (21.29) | | |
| **Modern contraceptive use** | | | | | |
| Ever used | 502 (37.04) | 853 (62.96) | 1,355 (53.20) | | <0.001 |
| Never used | 657 (55.13) | 535 (44.87) | 1,192 (46.80) | | |
| **Ever deliver in the health facility** | | | | | |
| Yes | 248 (23.34) | 813 (76.66) | 1,061 (51.59) | | <0.001 |
| No | 794 (79.83 | 201 (20.17) | 995 (48.41) | | |
| **Had at least one ANC\*** | | | | | |
| Yes | 818 (39.67) | 1,244 (60.33) | 2,062 (80.98) | | <0.001 |
| No | 341 (70.37) | 143 (29.63) | 484 (19.02) | | |
| **Current pregnancy desired** | | | | | |
| Then | 500 (40.31) | 740 (59.69) | 1,239 (48.66) | | <0.001 |
| Later | 276 (52.19) | 253 (47.81) | 529 (20.78) | | |
| Not at all | 104 (57.55) | 77 (42.45) | 181 (7.13) | | |
| No response | 279 (46.71) | 318 (53.29 | 597 (23.44) | | |

Note

\*ANC, Antenatal Care.

contraceptives, ever using facility delivery service before the recent birth, and attending at least one ANC follow-up were the individual-level factors that significantly increase the odds of institutional delivery. In addition, living in urban, having a partner who encourages ANC visits and living in a community where traditional birth attendats are not encouraged were among the household and community level factors which were significant factors which increase institutional delivery. Among the four domains of childbirth readiness index, availability of medicine and commodities in the nearby public health facility was significantly associated with an increased level of institutional delivery service utilization.

Based on this, when compared with women whose age were below 20 years, those who were between 20–34 years [AOR; 0.55 (95% CI; 0.32–0.85)] had 45% less chance of using

**Table 2. Community & household factors by place of delivery, PMA-Ethiopia 2019.**

| *Community & household* factors | Place of delivery | | | p-value |
|---|---|---|---|---|
| | **Home (976)** | **Facility (1571)** | **Total** | |
| | **N<u>o</u> (%)** | **N<u>o</u> (%)** | **N<u>o</u> (%)** | |
| **Residence** | | | | |
| Urban | 46 (7.98) | 534 (92.02) | 581 (22.81) | <0.001 |
| Rural | 1,113 (56.59) | 853 (43.41) | 1,966 (77.19) | |
| **Wealth Index** | | | | |
| Highest | 28 (5.43) | 484 (94.57) | 511 (20.08) | <0.001 |
| Second highest | 201 (39.20) | 312 (60.80) | 513 (20.13) | |
| Medium | 256 (50.01) | 256 (49.99) | 512 (20.08) | |
| Low | 308 (61.24) | 195 (38.76) | 507 (19.92) | |
| Very low | 366 (72.12) | 141 (27.88) | 504 (19.77) | |
| **Family size** | | | | |
| ≤ 3 | 250 (31.05) | 555 (68.95) | 804 (31.58) | <0.001 |
| 4–5 | 411 (46.80) | 467 (53.20) | 878 (34.47) | |
| ≥6 | 498 (57.65) | 366 (42.35) | 864 (33.94) | |
| **Community encourages facility delivery** | | | | |
| No/Don't know | 270 (75.02) | 90 (24.98) | 360 (14.14) | <0.001 |
| Most people | 393 (30.13) | 911 (69.87) | 1,304 (51.21) | |
| Some people | 250 (50.85) | 242 (49.15) | 493 (19.35) | |
| Few people | 245 (62.92) | 144 (37.08) | 390 (15.30) | |
| **Community encourages delivery with TBA** | | | | |
| No/Don't know | 335 (30.64) | 758 (69.36) | 1,093 (42.91) | <0.001 |
| Most people | 300 (65.79) | 156 (34.21) | 457 (17.94) | |
| Some people | 206 (56.16) | 161 (43.84) | 366 (14.38) | |
| Few people | 318 (50.39) | 313 (49.61) | 631 (24.77) | |
| **Community encourages ANC** | | | | |
| No/Don't know | 232 (70.51) | 97 (29.49) | 329 (12.94) | <0.001 |
| Most people | 386 (30.97) | 861 (69.03) | 1,247 (48.95) | |
| Some people | 282 (50.05) | 281 (49.95) | 563 (22.10) | |
| Few people | 259 (63.46) | 149 (36.54) | 408 (16.02) | |
| **Community encourages PNC** | | | | |
| No/Don't know | 317 (62.65) | 189 (37.35) | 505 (19.84) | <0.001 |
| Most people | 308 (30.57) | 699 (69.43) | 1,006 (39.51) | |
| Some people | 256 (46.20) | 299 (53.80) | 555 (21.80) | |
| Few people | 278 (57.96) | 202 (42.04) | 480 (18.84) | |

Note: ANC, Antenatal Care; PNC, Postnatal Care; TBA, Traditional Birth Attendant.

institutional delivery. Compared with women who attended more than secondary level education those who has no formal education[AOR: 0.19 (95% CI: 10.05–0.76)] or attended only primary level of education [AOR: 0.20 (95% CI: 0.05–0.75)] had decreased odd of institutional delivery. Also, attending at least one ANC visit increase the odds of institutional delivery by three folds [AOR: 3.09 (95% CI: 1.87–5.10)]. Similarly, the odds of institutional delivery will increase for women who ever used contraceptive methods [AOR; 3.09 (95% CI: 1.87–5.10)] and for women who ever gave birth in health facilities before the recent birth [AOR; 5.73 (95% CI; 4.00–8.19)] than their counterparts (Table 4).

Urban residents were about 11 times more likely to deliver in a health facility [AOR; 11.39 (95% CI; 5.56–23.31)] than their rural counterparts. Similarly, women whose partners didn't

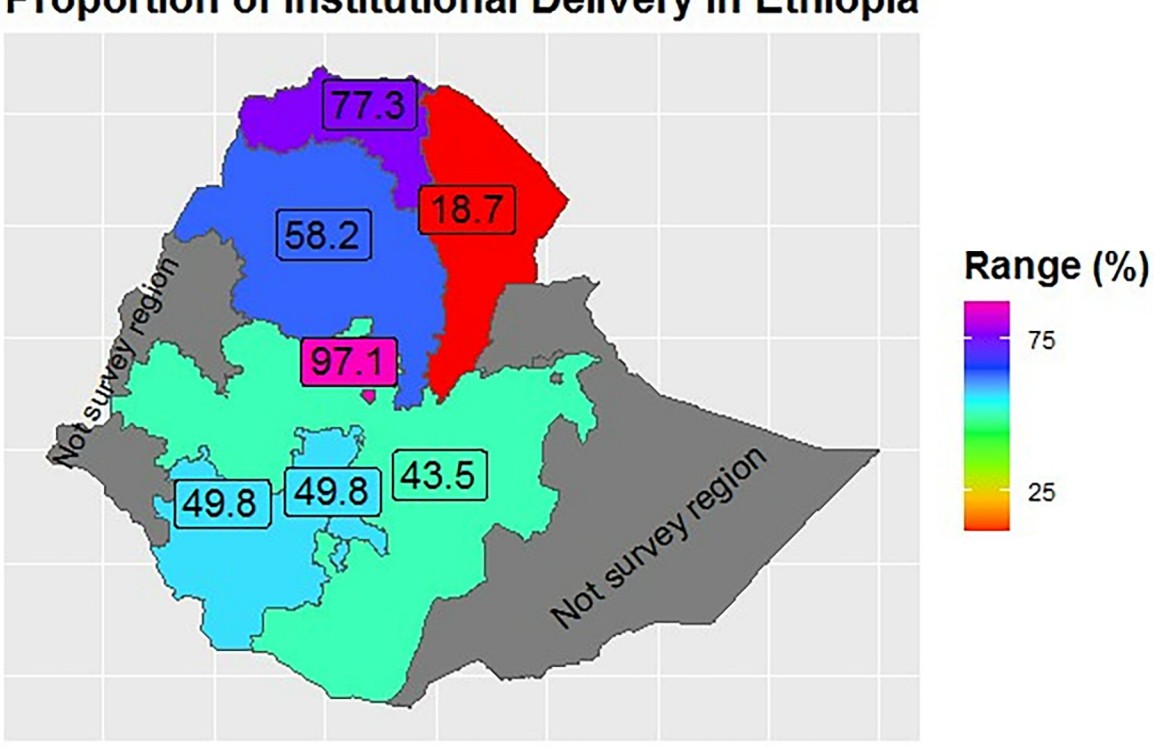

**Fig 1. Proportion of institutional delivery at different regions in Ethiopia.**

encourage utilization of ANC service [AOR; 0.57 (95% CI; 0.33–0.98)] have about 40% less chance of delivering in health facilities than their counterparts. Living in a community where traditional birth attendants were discouraged also increase the level of institutional delivery by 45% [AOR; 1.45 (95% CI; 0.91–2.31)]. In addition, when the readiness score for the availability of medicines and commodities in the closest facility increases, the odds of institutional delivery for the woman living close to the facility will increase by about 17 folds [AOR: 17.33 (95% CI; 1.32–26.4)] (Table 5).

## Discussion

In this study, we assessed the coverage and factors associated with institutional delivery among Ethiopian women. We found that, nearly half of pregnant women in Ethiopia gave birth outside health facilities. Our result also showed that younger maternal age, higher level of maternal education, ever using modern contraceptives, ever using facility delivery before the recent birth, and attending at least one ANC follow-up were the individual-level factors that significantly increase the odds of institutional delivery. In addition, living in urban, having a partner who encourages ANC visits and living in a community where traditional birth attendats are not encouraged were among the household and community level factors which were significant factors which increase institutional delivery. Among facility level factors, availability of medicine and commodities in the nearby public health facility was significantly associated with an increased level of institutional delivery service utilization.

In this study 54.49% of mothers had health facility delivery. Similarly, the study in Nepal also found that 53% pregenant women attend health facilities for delivery. However, a

**Table 3. WHO standards childbirth readiness index by domains of indicators and regions.**

| Categories reediness indicators and regions | Domain score (Mean score of items as a percentage) | | |
|---|---|---|---|
| | **Hospital** | **Health center** | **Both level** |
| **Medicines & commodities** | **35.58(±9.12)** | **31.84(±10.88)** | **32.61(±10.65)** |
| Tigray | 35.26(±10.01) | 35.89(±9.95) | |
| Afar | 30.13(±9.86) | 29.48(±6.92) | |
| Amhara | 45.21(±8.19) | **34.03(±8.04)** | |
| Oromia | 34.86(±3.86) | 29.83(±12.91) | |
| SNNPR | 35.11(±8.63) | 27.04(±10.93) | |
| Addis Ababa | 29.41(±1.00) | 38.47(±6.84) | |
| **Equipment, supplies, & amenities** | **85.94(±15.24)** | **75.26(±17.24)** | **77.46(±17.39)** |
| Tigray | 71.84(±17.36) | 79.44(±10.92) | |
| Afar | 97.07(±5.59) | 72.89(±12.80) | |
| Amhara | 90.85(±11.70) | 82.34(±11.90) | |
| Oromia | 93.68(±5.61) | 69.97(±15.43) | |
| SNNPR | 90.90(±8.56) | 64.21(±20.56) | |
| Addis Ababa | 93.33(±1.00) | 92.26(±9.28) | |
| **Performance of signal functions** | **84.84(±23.98)** | **60.64(±21.90)** | **65.63(±5.95)** |
| Tigray | 68.79(±34.99) | 54.32(±25.50) | |
| Afar | 100(±0.00) | 66.12(±20.83) | |
| Amhara | 8059(±13.62) | 64.99(±23.00) | |
| Oromia | 97.05(±4.58) | 54.83(±22.08) | |
| SNNPR | 90.50(±9.51) | 59.66(±17.61) | |
| Addis Ababa | 100(±0.00) | 71.61(±15.37) | |
| **Staffing & systems to support quality** | **90.75(±121.03)** | **84.93(±12.56)** | **86.13(±12.48)** |
| Tigray | 90.79(±08.00) | 92.68(±6.72) | |
| Afar | 92.99(±6.28) | 89.00(±10.70) | |
| Amhara | 80.05(±12.4) | 86.99(±13.31) | |
| Oromia | 96.84(±7.03) | 81.34(±11.79) | |
| SNNPR | 89.30(±13.27) | 78.70(±14.06) | |
| Addis Ababa | 100(±0.00) | 89.04(±7.33) | |
| **Overall weighted readiness** | **74.28(±9.95)** | **63.17 (±11.40)** | **65.46(±10.61)** |
| Tigray | 66.66(±13.20) | 65.58(±8.91) | |
| Afar | 80.05(±4.80) | 64.37(±9.54) | |
| Amhara | 74.17(±7.13) | 67.09(±7.63) | |
| Oromia | 80.61(±3.32) | 59.00(±10.59) | |
| SNNPR | 76.45(±4.21) | 57.40(±11.06) | |
| Addis Ababa | 80.69(±1.00) | 72.85(±4.03) | |

Note: SNNPR, South Nations Nationalities and Peoples Region.

multicenter study conducted in Bangladesh (60%), Tanzania (65%), Malawi (93%) and Senegal (77%) revealed higher coverage of facility delivery [29]. This variability could be explained by the differences in health care coverage and density of health care profesionals in these countries and Ethiopia. Based on the evidenced from WHO report on the ratio of health care providers to the total population, Ethiopia had small number of care providers in relation to the population size [30]. In contrast, facility birth in our study was higher than 39% reported in Haiti. This might be related to the poor health service delivery in Haiti due to the impact of frequent natural disasters on health infrastructures in the country [31].

**Table 4. Maternal & household factors associated with place of delivery, PMA-Ethiopia 2019.**

| Factors | Place of delivery | | AOR (95%CI) |
|---|---|---|---|
| | Facility No (%) | Home No (%) | |
| **Mothers' age** | | | |
| <20 years | 271(60.80) | 175(39.20) | 1 |
| 20–34 years | 937(55.32) | 756(44.68) | **0.55(0.32–0.85)** |
| 35–49 years | 180(44.15) | 228(55.85) | 0.58(0.31–1.08) |
| **Marital status** | | | |
| Married | 1,314(54.10) | 1,115(45.90) | 1 |
| Living together | 43(66.30) | 22(33.70) | 1.68(0.49–5.79) |
| Divorced/Widowed | 23(56.48) | 18(43.52) | 1.16(0.22–5.99) |
| Never Married | 8(63.25) | 5(36.75) | 1.47(0.13–16.27) |
| **Maternal education** | | | |
| Never attended | 395(38.11) | 642(61.89) | **0.19(0.05–0.76)** |
| Primary | 573(55.96) | 451(44.04) | **0.20(0.05–0.75)** |
| Secondary | 243(80.86) | 57(19.14) | 0.28(0.07–1.18) |
| More than secondary | 176(54.49) | 8(4.55) | **1** |
| **Modern contraceptive use** | | | |
| Ever used | 853(62.96) | 502(37.04) | **1.86(1.31–2.64)** |
| Never used | 535(44.87) | 657(55.13) | 1 |
| **Ever deliver in a health facility** | | | |
| Yes | 813(76.66) | 248(23.34) | **5.73(4.00–8.19)** |
| No | 201(20.17) | 794(79.83 | 1 |
| **Had at least one ANC** | | | |
| Yes | 1,244(60.33) | 818(39.67) | **3.09(1.87–5.10)** |
| No | 143(29.63) | 341(70.37) | 1 |
| **Birth outcome of 1ˢᵗ newborn** | | | |
| Live birth | 1,362(54.30) | 1,146(45.70) | 1 |
| Still birth | 26(66.85) | 13(0.51) | 1.90(0.20–17.96) |
| **Family size** | | | |
| ≤ 3 | 555(68.95) | 250(31.05) | 1.13(0.66–1.97) |
| 4–5 | 467(53.20) | 411(46.80) | 0.95(0.66–1.35) |
| ≥6 | 366(42.35) | 498(57.65) | 1 |

Note: ANC, Antenatal Care; AOR, Adjusted Odds Ratio; Those written in bold implies significant association.

Based on EDHS 2019, the proportion of mothers who had facility delivery was 70.4% in urban areas and 40.0% in rural. The overall proportion of facility delivery was 47.5% with regional variation from 94.8% in Addis Ababa to 28.3% in Afar [7]. The overall coverage and the variation among regions reported by EDHS 2019 were in line with our finding. Recent studies conducted in different parts of Ethiopia also reported that coverage of institutional delivery ranges from 27% to 51%. Among these studies, the lowest coverage of facility delivery (26.9%) was reported in Hadiya zone, Southern Ethiopia, whereas, 51.1% was reported in North West Ethiopia [8–13]. Most of these reports were lower than the overall proportion of facility delivery in our study. This might be explained by the geographic extent that these studies tried to address. Unlike our study that includes enumeration areas in big cities and remote rural settings, the other studies were done only in specific areas. As a result of higher institutional delivery coverage in big cities like Addis Ababa, the overall coverage in our study might be higher than the result from the other studies.

**Table 5. Community and health facility related factors associated with place of delivery, PM-Ethiopia 2019.**

| Factors | Place of delivery | | AOR |
|---|---|---|---|
| | Facility No (%) | Home No (%) | (95%CI) |
| **Residence** | | | |
| Urban | 534(92.02) | 46(7.98) | **11.39(5.56–23.31)** |
| Rural | 853(43.41) | 1,113(56.59) | **1** |
| **Partner encourage ANC** | | | |
| Yes | 1297(60.7) | 838(39.27) | 1 |
| No | 75(19.78) | 305(80.22) | **0.57(0.33–0.98)** |
| Has no partner | 16(51.28) | 15(48.72) | 1.19(0.19–7.10) |
| **Community encourages facility birth, ANC, PNC** | | | |
| Low | 706(63.22) | 410(36.78) | 1 |
| Medium | 266(51.10) | 255(48.90) | 0.94(0.56–1.60) |
| High | 416(45.73) | 493(54.27) | 1.17(0.77–1.78) |
| **Community encourages delivery with TBA** | | | |
| No/Don't know | 758(69.36) | 335(30.64) | **1.45(0.91–2.31)** |
| Most people | 156(34.21) | 300(65.79) | 0.96(0.58–1.61) |
| Some people | 161(43.84) | 206(56.16) | 0.89(0.54–1.46) |
| Few people | 313(49.61) | 318(50.39) | 1 |
| **Level of Health facility** | | | |
| Hospital | 258(62.26) | 156(37.74) | 1 |
| Health center | 1,130(52.98) | 1,002(47.02) | 0.92(0.45–1.88) |
| **Medicines & commodities Mean (±SD)** | 34.41 (±10.25) | 29.73 (±10.66) | **17.33(1.32–26.4)** |
| **Equipment, supplies, & amenities Mean (±SD)** | 79.79(±16.76) | 73.71(±17.73) | 0.32(0.05–2.06) |
| **Performance of signal functions Mean (±SD)** | 67.75(±24.46) | 62.22(±23.90) | 1.11(0.26–4.69 |
| **Staffing & quality systems Mean (±SD)** | 87.21(±11.93) | 84.37(±13.14) | 1.89(0.29–3.54) |

Note: ANC, Antenatal Care; AOR, Adjusted Odds Ratio; PNC, Postnatal Care; SD, Standard Deviation;TBA, Traditional Birth Attendant; No, Number; Those written in bold implies significant association.

In line with our finding, the study conducted by linking EDHS and SPA data found that rural residence was negatively associated with facility births [13]. A meta-analysis conducted on factors associated with institutional delivery in Ethiopia also found that women living in urban areas were more likely to have facility births compared with rural women [32]. This could be due to the fact that majority of the facilities in Ethiopia are concentrated in urban areas as well as the difference in educational level and access to information.

We also found that older age women were less likely to use institutional delivery. This is inline with other studies. Since older age women are in a traditional cohort, they are more likely to prefer home delivery [16]. However, our finding contradict with the fact that increased risk of obstetric complications and advers pregenancy outcomes with advanced maternal age that could alert the mother to have a continuum of care [33,34].

With regards to Women's educational level, our result revealed that when compared with women who attended more than secondary level education those who has no formal education or attended only primary level of education had decreased odd of institutional delivery. In con-sistent with our finding, both the special and the meta-analysis conducted in Ethiopia reported similar findings [13,32]. This implies that educated women have better information or

knowledge about maternal health services and better control over resources, which could all improve the rate of facility births [35].

Women who had at least one ANC visit were more likely to use institutional delivery service in our study. This result was in agreement with the results reported by a similar study conducted by linking EDHS and SPA data as well as the finding of the meta-analysis conducted to determine the effect of ANC on the use of institutional delivery in Ethiopia [13,36]. Other studies in Ethiopia also reported that the number of ANC visits was associated with institutional delivery [8,10,37]. In Ethiopia, only 43% of pregnant women have four or more ANC visits, and promoting women's use of ANC services should be emphasized to improve the utilization of health facility delivery services. In addition, ANC attendance can be used as a good opportunity to encourage utilization of facility delivery service [7].

Similarly, our finding indicates that, women whose partners encourage utilization of ANC service were more likely have institutional delivery than their counterparts. The finding from demographic and health survey of 28 developing countries also showed that rejection of maternal health services by partners was the most frequent reason given by women for not delivering in a facility as the man is the decision maker in the household. In addition, others studies also demonstrated positive impact of partner involvement in maternal health service utilization [38–41].

With regards to previous delivery place, our study showed that the odds having institutional delivery was higher among women who ever had facility delivered before the recent birth. Other studies also reported a finding that supports this result [42–44]. These consistant findings can be explained by the fact that women who had facility delivery in the past have already demonstrated some acceptance of the service. This might have impacted the subsequent health service use by avoiding fears and misconsumptions related with institutional delivery [42,45].

The other important predictor for utilization of institutional delivery was ever using contraceptive methods. Similar findings were reported by other studies conducted elsewhere [46,47]. This association can be explained by the fact the family planning is considered a key intervention to achieve almost all global development agenda. Family planning use could also reflect the woman's concern of her pregnancy and her maternal health service acceptance level [48].

Regarding health facility level factors, we found that when the readiness score for the availability of medicines and commodities in the closest facility increases, the odds of using institutional delivery service by the woman living close to the facility will increase. Similarly, the study conducted in Tanzania found that the availability of tracer drugs has moderate association with births in a health facility [49]. However, the other three domains we used to measure the delivery service readiness score of the facility closest to women's house (Equipment, supplies, and amenities available; Performance of signal functions indicators and Staffing and systems to support quality in the facilities) were not significantly associated with the probability of facility delivery by the woman. This indicates, that different dimensions of facility readiness measurements have different contributions in promoting institutional delivery in the catchment.

## Strength and limitation

Our analysis had several strengths which can address limitations that hinder current research efforts as well as previous studies conducted by linking different data sources. The main strength of the current study is the use of panel household survey and health facility data collected simultaneously by PMA-Ethiopia. Previous studies were conducted by linking EDHS and SPA data. As a result, the time and location difference between the individual data from

DHS and facility data from SPA limits the inferences that can be made from the linked data analysis [13,19]. In addition to the survey protocol used to select all public SDPs that were administratively assigned to serve the 265 EAs, we also applied geographically closest air distance for link households and facilities. Furthermore, we linked individuals with the closest SDP because linking at the stratum level will result in large variations in the measurement of health facility readiness between facilities in the same stratum; which will affect the reliability of estimates. Our analysis was also based on the well-established concept that access to healthcare is a product of the quality of nearby facilities and their distance [22]. The other advantage of using this data source is that it provides a nationally representative sample of pregnant women and health facilities in Ethiopia. The indicators used to measure facility readiness for delivery service were comprehensive as it was selected if the item was recommended by either WHO-SARA [25]. or by the Newborn Indicator Technical Working Group (Save the Children Federation, Inc. 2017), or by Gabrysch's New Signal Functions [16].

However, this study had some limitations. The first weakness is the fact that the spatial data from PMA-Ethiopia are randomly displaced as a confidentiality measure. Second, linking mothers with their actual place of delivery was not possible due to the unavailability of the information in the survey data set. Third, knowledge, attitude, and practice of health care providers were not assessed as they might have a contribution to the quality of the service.

## Conclusion

The finding of this study revealed that nearly half of the total deliveries in Ethiopia took place outside health facilities. The study also showed wide geographical variations of institutional delivery and strongly significant differences between rural and urban residents. Younger maternal age, higher level of maternal education, partner involvement in encouraging maternal service utilization, having at least one ANC visits and living in a community where traditional birth attendants are descourged were predictors of institutional delivery in Ethiopia. This implies the need to address health service inequality in the country and emphasize the importance of improving ANC attendance to increase the rate of health facility delivery. Interventions to increase utilization of other maternal health services by the woman and involvement of their partners and communities should be part of the strategies to improve effective coverage of health facility delivery. Facility level factors should also be considered as one of the key intervention areas to improve institutional deliver. Based on this, it is also important to improve the availability of medicine and commodities in the nearby public hospitals and health centers.

## Supporting information

**S1 Table. Description of indicators used to measure facility readiness for delivery service.** (DOCX)

## Acknowledgments

We would like to acknowledge members of the PMA project in Ethiopia and United state. Special thanks for Saifuddin Ahmed, Sally Safi, Selam Desta, Shannon Wood, Mahari Yihdego and Tesfamichael Awoke. We are grateful for technical and financial support from Bill & Melinda Gates Institute for Population and Reproductive Health at the Johns Hopkins Bloomberg School of Public Health. The JHU Bloomberg School of Public Health also provided us with the data.

## Author Contributions

**Conceptualization:** Fanuel Belayneh Bekele, Adiam Nega, Assefa Seme, Solomon Shiferaw.

**Data curation:** Fanuel Belayneh Bekele.

**Formal analysis:** Fanuel Belayneh Bekele, Kasiye Shiferaw, Adiam Nega, Anagaw Derseh, Assefa Seme, Solomon Shiferaw.

**Funding acquisition:** Assefa Seme, Solomon Shiferaw.

**Investigation:** Fanuel Belayneh Bekele, Kasiye Shiferaw, Anagaw Derseh, Assefa Seme, Solomon Shiferaw.

**Methodology:** Fanuel Belayneh Bekele, Kasiye Shiferaw, Adiam Nega, Anagaw Derseh, Assefa Seme, Solomon Shiferaw.

**Project administration:** Fanuel Belayneh Bekele, Assefa Seme, Solomon Shiferaw.

**Resources:** Assefa Seme, Solomon Shiferaw.

**Software:** Fanuel Belayneh Bekele, Kasiye Shiferaw, Adiam Nega, Anagaw Derseh, Assefa Seme, Solomon Shiferaw.

**Supervision:** Fanuel Belayneh Bekele, Assefa Seme, Solomon Shiferaw.

**Validation:** Fanuel Belayneh Bekele, Assefa Seme, Solomon Shiferaw.

**Visualization:** Fanuel Belayneh Bekele.

**Writing – original draft:** Fanuel Belayneh Bekele.

**Writing – review & editing:** Kasiye Shiferaw, Adiam Nega, Anagaw Derseh, Assefa Seme, Solomon Shiferaw.

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
