## [Decision Letter · Decision Letter 0]

16 Jun 2022

Factors influencing place of delivery in Ethiopia: Linking individual, household, and health facility-level data

PGPH-D-22-00394

Dear Bekele,

We are pleased to inform you that your manuscript 'Factors influencing place of delivery in Ethiopia: Linking individual, household, and health facility-level data' has been provisionally accepted for publication in PLOS Global Public Health.

Best regards,

Collins Otieno Asweto, PhD

Academic Editor

Reviewer's Responses to Questions

**Comments to the Author**

1. Does this manuscript meet PLOS Global Public Health’s publication criteria? Is the manuscript technically sound, and do the data support the conclusions? The manuscript must describe methodologically and ethically rigorous research with conclusions that are appropriately drawn based on the data presented.

Reviewer #1: Yes

Reviewer #2: No

2. Has the statistical analysis been performed appropriately and rigorously?

Reviewer #1: Yes

Reviewer #2: No

3. Have the authors made all data underlying the findings in their manuscript fully available (please refer to the Data Availability Statement at the start of the manuscript PDF file)?

Reviewer #1: Yes

Reviewer #2: No

4. Is the manuscript presented in an intelligible fashion and written in standard English?

Reviewer #1: Yes

Reviewer #2: No

5. Review Comments to the Author

Reviewer #1: They have done a good job regarding delivery place in Ethiopia. The number of individuals in the data was also respectful. For analysis, they have used some descriptive statistics and logistic regression which was acceptable too.

Reviewer #2: This manuscript does not meet the standards of an international journal. First off - there are sections where the writing is nearly unintelligible and has significant errors - for example, see lines 234-241. The dataset needs to be described in greater details - especially the panel data collection strategy for individual women and the type of data that is collected. The analytical approach section is incomplete as is the Results The Results section also does not follow convention and the results are hard to understand - there are too many tables and the main results needs to be better presented. The Discussion section is just a continuation of the Results section. Since there are significant gaps in nearly every section, I have decided to reject this paper.

6. PLOS authors have the option to publish the peer review history of their article (what does this mean?). If published, this will include your full peer review and any attached files.

**Do you want your identity to be public for this peer review?** For information about this choice, including consent withdrawal, please see our Privacy Policy.

Reviewer #1: No

Reviewer #2: No
